# Establishing multiple omics baselines for three Southeast Asian populations in the Singapore Integrative Omics Study

Woei-Yuh Saw[1,2], Erwin Tantoso[1], Husna Begum[2,3], Lihan Zhou[4], Ruiyang Zou[4], Cheng He[4], Sze Ling Chan[5], Linda Wei-Lin Tan[1], Lai-Ping Wong[1], Wenting Xu[1], Don Kyin Nwe Moong[1], Yenly Lim[1], Bowen Li[1], Nisha Esakimuthu Pillai[2], Trevor A. Peterson[6,7], Tomasz Bielawny[6,7], Peter J. Meikle [3,8], Piyushkumar A. Mundra[3], Wei-Yen Lim[1], Ma Luo[6,7], Kee-Seng Chia[1], Rick Twee-Hee Ong[1], Liam R. Brunham[5], Chiea-Chuen Khor [9,10], Heng Phon Too[11,12,13], Richie Soong[14], Markus R. Wenk[2,11,15,16,17], Peter Little [2] & Yik-Ying Teo[1,2,9,15,18]

The Singapore Integrative Omics Study provides valuable insights on establishing population reference measurement in 364 Chinese, Malay, and Indian individuals. These measurements include > 2.5 millions genetic variants, 21,649 transcripts expression, 282 lipid species quantification, and 284 clinical, lifestyle, and dietary variables. This concept paper introduces the depth of the data resource, and investigates the extent of ethnic variation at these omics and non-omics biomarkers. It is evident that there are specific biomarkers in each of these platforms to differentiate between the ethnicities, and intra-population analyses suggest that Chinese and Indians are the most biologically homogeneous and heterogeneous, respectively, of the three groups. Consistent patterns of correlations between lipid species also suggest the possibility of lipid tagging to simplify future lipidomics assays. The Singapore Integrative Omics Study is expected to allow the characterization of intra-omic and inter-omic correlations within and across all three ethnic groups through a systems biology approach.

[1] Saw Swee Hock School of Public Health, National University of Singapore, 12 Science Drive, Singapore 117549, Singapore. [2] Life Sciences Institute, National University of Singapore, 28 Medical Drive, Singapore 117456, Singapore. [3] Baker IDI Heart and Diabetes Institute, 75 Commercial Road, Melbourne, VIC 3004, Australia. [4] MiRXES, Agency for Science, Technology and Research Singapore, 10 Biopolis Road, Chromos, Singapore 138670, Singapore. [5] Translational Laboratory in Genetic Medicine, Agency for Science, Technology and Research Singapore, 8A Biomedical Grove, Immunos, Singapore 138648, Singapore. [6] Department of Medical Microbiology, University of Manitoba, 730 William Avenue, Winnipeg, MB, Canada R3E 0Z2. [7] National Microbiology Laboratory, 1015 Arlington St, Winnipeg, MB, Canada R3E. [8] Department of Biochemistry and Molecular Biology, The University of Melbourne, Bio21, 30 Flemington Road, Melbourne, VIC 3010, Australia. [9] Genome Institute of Singapore, Agency for Science, Technology and Research Singapore, 60 Biopolis St, Singapore 138672, Singapore. [10] Singapore Eye Research Institute, 20 College Road, Singapore 169856, Singapore. [11] Department of Biochemistry, Yong Loo Lin School of Medicine, National University of Singapore, 8 Medical Drive, Singapore 117597, Singapore. [12] Molecular Engineering of Biological and Chemical System/Chemical Pharmaceutical Engineering, Singapore-Massachusetts Institute of Technology Alliance, 4 Engineering Drive 3, Singapore 117576, Singapore. [13] Bioprocessing Technology Institute, A*STAR (Agency for Science, Technology and Research, Singapore), 20 Biopolis Way, Singapore 138668, Singapore. [14] Cancer Science Institute of Singapore, National University of Singapore, 14 Medical Drive, Singapore 117599, Singapore. [15] NUS Graduate School for Integrative Science and Engineering, National University of Singapore, 28 Medical Drive, Singapore 117456, Singapore. [16] State Key Laboratory of Molecular Developmental Biology, Institute of Genetics and Developmental Biology, Chinese Academy of Sciences, No.1 West Beichen Road, Chaoyang District, Beijing 100101, China. [17] Department of Biological Sciences, National University of Singapore, 16 Science Drive 4, Singapore 117543, Singapore. [18] Department of Statistics and Applied Probability, National University of Singapore, 6 Science Drive 2, Singapore 117546, Singapore. Correspondence and requests for materials should be addressed to Y.-Y.T. (email: statyy@nus.edu.sg).

Knowledge of the genetic determinants of common human diseases has increased tremendously in the past decade, mostly from discoveries made by genome-wide association studies (GWAS)[1–3]. The efficient design of GWAS for querying the entire genome benefitted from the arrival of the HapMap resource, which produced a genomic map that outlined the correlation patterns in the human genome for identifying tagging single-nucleotide polymorphisms (SNPs)[4, 5]. The HapMap resource also provided a public database on how prevalent specific alleles are in different ancestry groups in the world[6]. The subsequent development of national genome variation projects has thus produced numerous public databases that have been instrumental at enabling genetics as a forerunner in precision medicine[7–10]. For instance, the predecessor of Singapore Integrative Omics Study (iOmics), the Singapore Genome Variation Project[7], which only focused on making static genetic SNP and Human Leukocyte Antigen (HLA) measurements, indeed facilitated numerous investigations into the population genetics and genetics of common diseases in Asian communities, while at the same time allowing cost-effectiveness assessments and burden estimation of pharmacogenetic testing prior to initiate drug treatments[11, 12], which consequentially influenced policies on governmental subsidies for the costs of genetic tests[13].

Technological advances have facilitated the measurement of biological states other than genetics, such as quantifying the extent of messenger RNA (mRNA) transcription by expression hybridization profiling[14–17] or in assessing the abundance of different lipid molecules with mass spectrometry[18–20]. When the transcriptome or lipidome of multiple individuals are measured, the expression or quantification of specific sub-units (whether gene or lipid molecule) can segregate between subgroups of individuals, rendering these segregating sub-units as effective biomarkers for the subgroupings[21–27], not unlike what is currently happening in GWAS. However, unlike the plethora of public genetic databases, there is presently an absence of systems-level maps to properly characterize the transcriptome and lipidome in the general population.

In this paper, we introduce the iOmics, which aims to establish population reference measurements across multiple omic technologies in three major populations in Singapore (see Table 1), and to interrogate the extent that different omic and lifestyle measurements differ between the three populations. The demography of Singapore is made up of three main ethnic communities comprising the Chinese, the Malays, and the Indians. The genetics of these populations has been previously systematically characterized by the Singapore Genome Variation Project[7, 28–30], which mapped the predominant genetic ancestries of these populations, respectively, to southern Han Chinese, a cosmopolitan admixture of Malays from Indonesia and Malaysia, and Tamil Indians from south India, respectively. In the present setup, measurements have been made in the iOmics to investigate the baseline genetics, transcription, lipid levels, and miRNAs expression. Each technology was selected for the purpose of evaluating the value of information in the biological cascade from DNA to RNA, and to biological units (cellular lipids) that are close surrogates to expressed phenotypes. The iOmics is expected to facilitate biomedical science experiments, investigating the impact of an omic measurement on biological processes or outcomes by interrogating the presence and extent of intra-omic and inter-omic correlation. In addition to the unprecedented collection of omic measurements made on the same individuals, the design and ethical set-up of the iOmics specifically offers the unique opportunity to recall participating subjects back for additional experiments according to the desired biological profiles. The data for the iOmics resource is publicly available at //phg.nus.edu.sg/#iomics.

## Results

**Quality control (QC) of samples and variants.** The iOmics surveyed 122 Chinese (72 females), 120 Malays (77 females), and 122 Indians (79 females) from the longitudinal Multi-ethnic Cohort of the Singapore Population Health Studies (SPHS) (https://www.sph.nus.edu.sg/research/sphs), with specific ethical approval and informed consent obtained for re-contacting the participants according to their biological and omic profiles. Each individual was genotyped on the Illumina 2.5M and Exome microarrays to yield a genetic resource of 2,527,458 SNPs, as well as on a customized pharmacogenetics microarray with 4032 SNPs after QC. Classical HLA alleles on all eight Class I and Class II loci were obtained from sequence-based allelotyping, yielding an allelic resolution of at least four digit for all the samples. Expression levels were available for 21,649 gene transcripts, and lipidomics profiling successfully measured the content of 282 unique lipid species. The normalized counts for 274 non-coding RNAs (miRNAs) were also measured. As all 364 subjects were

| | Details | Sample size per ethnicity (C/M/I) |
|---|---|---|
| **Table 1 Spectrum of omics and non-omics measurements available in the iOmics** | | |
| *Omics* | | |
| Genomics | ● Illumina 2.5M microarray genotyping | ● 110/108/105 |
| | ● Illumina exome chip genotyping | ● 110/108/105 |
| | ● Pharmacogenomics SNP typing (4032 SNPs) | ● 106/112/115 |
| | ● HLA typing (-A, -B, -C, -DPA, -DPB, -DQA, -DQB, -DRB) | ● 111/119/120 |
| | ● Deep (30×) whole-genome sequencing | ● 0/62/38 |
| Lipidomics | ● Mass spectrometry with Multiple Reaction Monitoring of 282 lipid molecules in three major lipid classes (glycerophospholipids, sphingolipids, sterols) | ● 122/117/120 |
| Transcriptomics | ● Affymetrix HumanGene 1.0 ST array | ● 98/75/96 |
| MicroRNA | ● mSMRT-qPCR miRNA assay of 274 circulating miRNAs | ● 117/115/119 |
| *Non-omics* | | |
| Nutrition | ● Validated interviewer-directed Food Frequency Questionnaire (199 dietary variables) | ● 122/116/120 |
| Lifestyle and environment | ● Interviewer-directed questionnaire, including smoking, alcohol consumption, and physical activity (46 lifestyle variables) | ● 122/116/120 |
| Clinical measurements | ● Clinically assessed measurements and assays, including age, sex, height, weight, BMI, HDLc, LDLc, TG, BP, total cholesterol, HbA1c, fasting glucose (39 clinical variables) | ● 122/116/120 |

*Note:* The sample sizes stated here refer to the number of subjects that remained after assessment for data quality

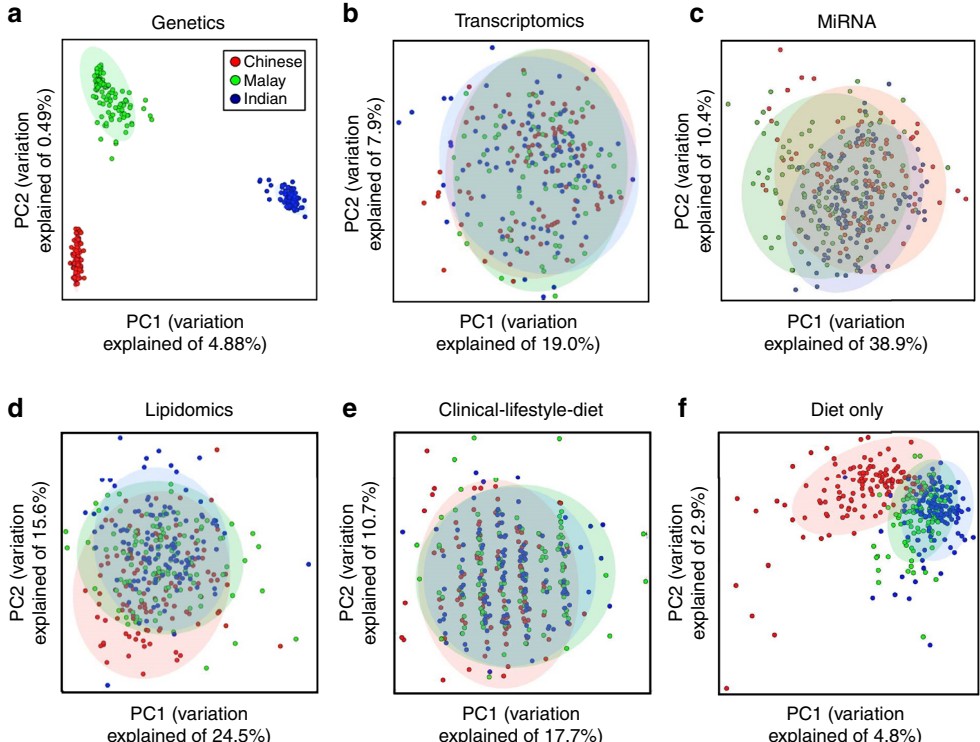

**Fig. 1** PCAs of omics and clinical/lifestyle/diet data. Biplots are shown for five distinct PCAs using the respective first two axes of variations from each PCA. The five PCAs correspond to the analysis of: **a** 101,099 autosomal SNPs pseudo-randomly chosen to minimize linkage disequilibrium between the SNPs; **b** 21,649 gene transcript probesets; **c** 274 miRNAs; **d** 282 lipid species; **e** a set of 284 clinical, lifestyle, and dietary variables; and **f** only the 199 dietary variables. Each *circle* represents an individual from the iOmics and is assigned a color corresponding to the self-reported ethnicity of the subject, according to the color legend on the top right panel in **a**

participants of an ongoing longitudinal cohort study, there were 296 non-omic variables that were related to clinical, lifestyle, and dietary indicators of each participant. A subset of the 364 individuals (62 Malays, 38 Indians) has also undergone deep whole-genome sequencing to a target depth of 30-fold coverage, although the resultant coverage was in excess of 40-fold for most subjects[31, 32] (Supplementary Table 1). The number of subjects that remained after QC differed for each omic platform, and details can be found in Supplementary Table 2.

**Principal component analyses (PCAs) of cryptic relatedness.**
A series of PCAs were performed in order to derive the extent of similarity between the subjects, using information from each of the four omic technologies and from the clinical, lifestyle, and dietary measurements. Unsurprisingly, the PCA with genetic data (101,099 SNPs) yielded distinct clusters corresponding to the self-reported ethnicities, with the first axis of variation distinguishing the Indians from the Chinese ($F_{ST}$ with Indians = 3.0%) and Malays ($F_{ST}$ with Indian = 2.0%), and the second axis separating the Chinese from the Malays ($F_{ST}$ = 1.0%, Fig. 1a). In contrast, the PCA with transcriptomics data (21,649 gene transcripts) and the 274 miRNAs did not yield any discernible separation between the ethnic groups (Fig. 1b, c). The PCA with the lipidomics data (282 lipid species) revealed marginal separation between the Chinese and non-Chinese on the second axis of variation, although it was not possible to separate between the three populations on the leading axis of variation (Fig. 1d). While the PCA using 284 non-omic clinical, lifestyle, and dietary variables did not yield any striking differentiation between the three ethnic groups (Fig. 1e), the PCA using only the 199 dietary variables was able to distinguish between the Chinese and

non-Chinese (Fig. 1f, Supplementary Fig. 1), despite the dietary variables being a smaller subset of the non-omic variables in the former analysis.

**Genetics.** The Wright $F_{ST}$ metric was used to quantify the extent of allele frequency difference at each SNP across the three ethnic groups, and we searched for contiguous stretches of the genome where there was an over-representation of SNPs with high $F_{ST}$ values. A total of 520 regions were identified to exhibit significant evidence of inter-ethnic difference (Supplementary Data 2), and all of these regions were driven by allele frequency differences between Indians and non-Indians. We observed 479 regions to be driven by frequency differences between Indians and Chinese, and the remaining 41 regions were driven by differences between Indians and Malays.

For the 4032 pharmacogenomic SNPs, we identified six SNPs that were differentiated between the ethnic groups (Table 2, Supplementary Fig. 2), including four tightly linked SNPs in *VKORC1* that have been established to correlate with optimal warfarin dosaging[33–35]. Similar to the genome-wide evidence seen previously, the differentiation of these six SNPs was most striking between Indians and Chinese. The remaining two SNPs were located in the alcohol dehydrogenase 4/5 genes (*ADH4/5*) and in the ATP-binding cassette sub-family B member 5 gene (*ABCB5*), respectively; the former are genes responsible for the metabolism of alcohol substrates, while the latter gene is involved in the development of drug (doxorubicin) resistance to melanoma treatment.

When we interrogated the extent of inter-ethnic variation at the 198 HLA alleles across 8 HLA loci, 20 alleles exhibited $F_{ST} \geq 0.05$ of which 12 were driven by frequency differences

**Table 2 Six candidate pharmacogenomic variants of most differentiated between three ethnic groups**

| SNP | CHR | POS | Alleles | Gene region | Frequency (Chinese) | Frequency (Malay) | Frequency (Indian) | Clinical PGx implication[a] | Wright $F_{ST}$ | $P_{empirical}$ |
|---|---|---|---|---|---|---|---|---|---|---|
| rs2359612 | 16 | 31011297 | A/G | VKORC1, intron | 0.118(G) | 0.263(G) | 0.900 (G) | (i) Patients with AA genotype who are treated with warfarin may require lowest dose as compared to patients with the AG or GG genotype (ii) Patients with the AG genotype who are treated with warfarin may require lower dose as compared to patients with the GG genotype (iii) Patients with the GG genotype who are treated with warfarin may require higher dose as compared to patients with the AG or AA genotype | 0.471 | 1.23E-05 |
| rs749671 | 16 | 30995848 | A/G | VKORC1 ZNF646, coding SYN | 0.118(G) | 0.273(G) | 0.900 (G) | NA | 0.466 | 1.31E-05 |
| rs8050894 | 16 | 31012010 | C/G | VKORC1, intron | 0.118(C) | 0.263 (C) | 0.868 (C) | (i) Patients with the CC genotype who are treated with warfarin may require a higher dose as compared to patients with the CG or GG genotype (ii) Patients with the CG genotype who are treated with warfarin may require a lower dose as compared to patients with the GG genotype (iii) Patients with the GG genotype who are treated with warfarin may require the lowest dose as compared to the patients with the CG or CC genotype | 0.434 | 2.33E-05 |
| rs7294 | 16 | 31009822 | C/T | VKORC1, flanking UTR | 0.109(T) | 0.272 (T) | 0.822(T) | (i) Patients with the CC genotype who are treated with warfarin may require a lower dose as compared to patients with the CT or TT genotype (ii) Patients with the CT genotype who are treated with warfarin may require a higher dose as compared to patients with the CC genotype (iii) Patients with the TT genotype who are treated with warfarin may require a higher dose as compared to patients with the CC genotype | 0.387 | 4.79E-05 |
| rs1238741 | 4 | 100202335 | C/T | ADH4/5, flanking UTR | 0.179 (T) | 0.272 (T) | 0.865 (T) | NA | 0.375 | 5.78E-05 |
| rs11974407 | 7 | 20695644 | C/G | ABCB5, intron | 0.038 (C) | 0.165 (C) | 0.670(C) | NA | 0.361 | 7.08E-05 |

[a]Information were retrieved from PharmGKB®, only clinical implication with level 2a, 2b, or 1 were retrieved

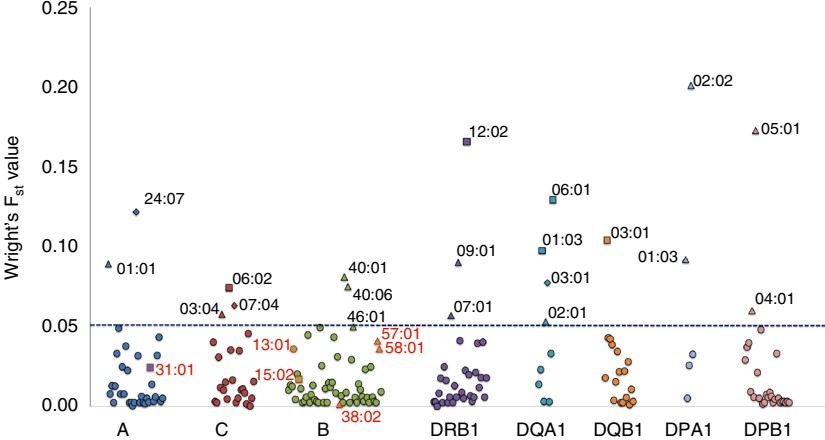

**Fig. 2** Distribution of the Wright $F_{ST}$ value across three ethnic groups at the eight HLA loci. The distribution of the Wright $F_{ST}$ value across three populations at the eight HLA loci. The alleles shown in the plot are the top three $F_{ST}$ alleles at each HLA loci. The *triangular* shape indicates the HLA alleles, where the $F_{ST}$ values are driven by differences between Chinese and Indians. The *diamond* shape indicates the HLA alleles, where the $F_{ST}$ values are driven by the differences between Chinese and Malays. The *square* shape indicates the HLA alleles, where the $F_{ST}$ values are driven by the differences between Malays and Indians. Shapes with *red* color outline are representing drug-associated HLA alleles[41] (Table 3)

| Table 3 Drug-associated HLA alleles | | | | | | |
|---|---|---|---|---|---|---|
| HLA allele | Drug | Adverse reaction | Allele frequency (%) | | | Wright $F_{ST}$ value |
| | | | **Chinese** | **Malay** | **Indian** | |
| A*31:01 | Carbamazepine | Rash | 0.9 | 0 | 5.0 | 0.025 |
| B*15:02 | Carbamazepine Phenytoin | SJS | 7.4 | 12.4 | 3.8 | 0.017 |
| B*13:01 | Dapsone | HSS | 10.2 | 3.0 | 0.8 | 0.036 |
| B*38:02 | Sulfomethoxazole | SJS/TEN | 4.6 | 4.3 | 2.9 | 0.001 |
| B*57:01 | Abacavir Flucloxacilin | HSS DILI | 0 | 0.9 | 7.5 | 0.041 |
| B*58:01 | Allopurinol | SJS | 10.2 | 3.0 | 0.8 | 0.036 |

*DILI* drug-induced liver injury, *HSS* hypersensitivity syndrome, *SJS* Stevens–Johnson syndrome, *TEN* toxic epidermal necrolysis

between Chinese and Indians, 5 by differences between Malays and Indians, and the remaining 3 by differences between Chinese and Malays (Fig. 2, Supplementary Table 3). The list of 20 alleles included B*40:01, which is present at a higher frequency in Chinese (20.8%) compared to the non-Chinese (4.3% in Malays, 2.5% in Indians), and the carriage of this allele has been linked to a decreased risk of carbamazepine-induced severe cutaneous adverse reactions such as Stevens-Johnson syndrome and toxic epidermal necrolysis[36, 37]. There were five other HLA alleles known to be pharmacogenetically important due to their strong associations with adverse drug responses, and we observed that B*38:02 exhibited low degree of differentiation in our populations, although the remaining four (A*31:01, B*15:02, B*57:01, B*58:01) exhibited modest degree of variation ($F_{ST} \geq 1.5\%$, Table 3).

**Transcription**. Of the 21,649 transcription probesets, 280 probesets were identified to be differentially expressed across the three ethnic groups, although the majority (276) were attributed to expression differences between Indians and non-Indians, especially against the Chinese (Supplementary Data 3). The three leading differentially expressed genes corresponded to: (i) Urotensin II (*UTS2*), where the levels of gene expression for Indians were almost three-fold lower in Chinese and Malays ($P_{Bonferroni} = 1.98 \times 10^{-24}$, Fig. 3a); (ii) Homo sapiens phospholipased B1 (*PLB1*) where Indians and Chinese exhibited the highest and lowest level of gene expression, respectively ($P_{Bonferroni} = 1.52 \times 10^{-13}$, Fig. 3b); and (iii) TRAF-interacting

protein with forkhead-associated domain (*TIFA*) where similarly Indians and Chinese presented the highest and lowest level of gene expression, respectively ($P_{Bonferroni} = 6.76 \times 10^{-12}$, Fig. 3c). Notably, we observed that *BRCA1* expression levels were different between Chinese and non-Chinese (Supplementary Fig. 3, $P_{Bonferroni} = 1.27 \times 10^{-3}$), and this intriguingly concurred with the trend that Singapore Chinese possessed an almost 11% higher age-standardized incidence rate for breast cancer compared to Singapore Indians (https://www.nrdo.gov.sg/publications/cancer, NRDO Singapore Cancer Registry Interim Report 2010–2014, accessed 18 August 2016). A functional enrichment pathway analysis of the 280 probesets against the DAVID 6.7 Biological Database[38] revealed that 44 probesets (16%) were significantly enriched in immune response and regulatory pathways ($P_{FDR} < 0.05$), and all of these 44 probesets were differentially expressed between Indians and non-Indians.

**miRNA**. Of the 274 miRNAs, we observed 5 miRNAs to be differentially expressed across the three ethnic groups, of which 4 were driven by expression differences between the Chinese and Malays and the remaining miRNA (*hsa-miR-375*) was driven by differences between Indians and non-Indians (Table 4, Supplementary Fig. 4A–E).

**Lipidomics**. The set of 282 lipid species came from 4 lipid categories (glycerophospholipids, sphingolipids, sterol lipids, glycerolipids), of which there were 20 lipid classes

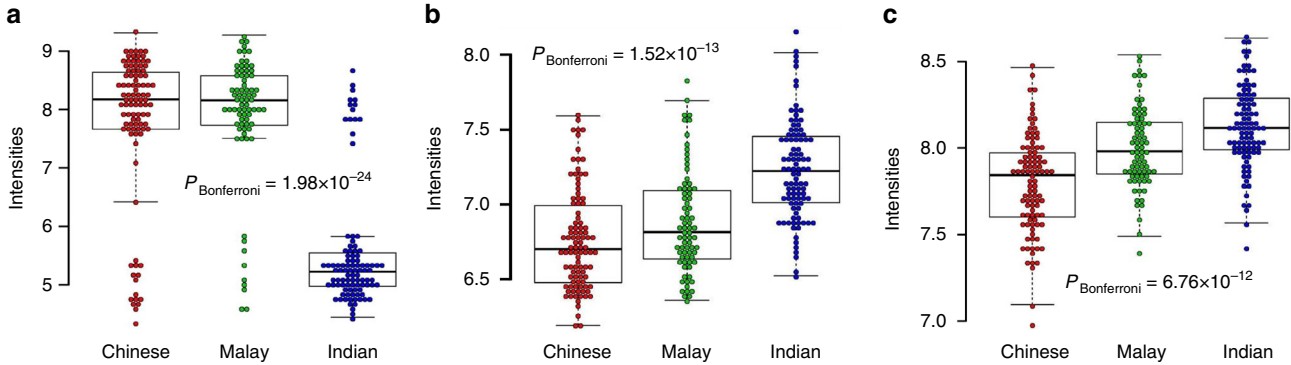

**Fig. 3** A combined boxplot and scatter plot of the top three significant transcript probesets across three populations. The combined plot showing distribution of transcript intensities of **a** 7912136:*UTS2* gene, **b** 8041061:*PLB1* gene, **c** 8102362:*TIFA* gene across three populations. *P*-values were calculated using ANOVA, adjusted for batch effect and gender, and corrected for Bonferroni. The upper whisker represents either the maximum value observed or is 1.5 times the interquartile range greater than the third quartile, whichever is smaller. The lower whisker represents either the minimum value observed or is 1.5 times lower than the first quartile, whichever is greater. The details of the significant transcript probesets across three populations can be found in Supplementary Data 3

(Supplementary Table 4). We identified 107 lipid species where the lipid levels were differentiated between the 3 ethnic groups, and there are 20 lipid species that were either 1.5 fold-change higher or lower in their lipid levels with respect to Chinese (Supplementary Data 4 and Supplementary Fig. 5). Of the 107 differentiated lipid species, 91 were the low abundant of the lipids on a molar basis (nmol/ml), whereas 16 were observed among the major abundant species with mean lipid concentration >10 nmol/ml (Supplementary Data 4). Majority (86 lipids) belonging to the glycerophospholipid category, which comprised the main component of the biological membrane, serving to (i) stabilize and isolate the intracellular environment from the external environment; (ii) regulate the transportation of the molecules through the membrane; and (iii) organize membrane component into localized areas involved in specific processes such as signal transduction[39]. Of the 107 differentiated lipid species, 88 were driven by lipid-level differences between Indians and non-Indians, and the remaining 19 were driven by differences between Chinese and Malays. In fact, the three most differentiated lipid species were due to differences seen between Indians and non-Indians (Supplementary Fig. 6A–C), at (i) PC(O-40:7) ($P_{Bonferroni} = 5.78 \times 10^{-24}$); (ii) PC 38:3 ($P_{Bonferroni} = 3.84 \times 10^{-21}$); and (iii) PE(O-40:7) ($P_{Bonferroni} = 3.54 \times 10^{-20}$).

Our analyses also suggested that lipid species with similar chemical properties that were categorized in the same lipid class have a tendency to be correlated. As a result, we investigated the extent of correlation between lipid molecules within each ethnic group, both within lipid class and between lipid classes in each of the three ethnic groups (Fig. 4). We identified 29 lipid species in the Chinese that can effectively represent the information from the measurements of 71 lipid species (defined as $r^2 > 0.8$). For the Malays, there were 26 tagging lipids for 61 species; and 26 tagging lipids for 60 species for the Indians (see Supplementary Tables 5–7). Notably, lipid classes in the sterol (free cholesterol (COH), cholesteryl ester (CE), and glycerolipids (diglycerides (DG) and triglycerides (TG))) categories were highly correlated within and between classes, with lipids from DG and TG accounting for 22% (28/125), 16% (21/125), and 3% (2/65) of the observed lipid pairings. This meant that the 282 lipid species can be summarized by 240, 247, and 248 'tagging' lipids in the Chinese, Malays, and Indians, respectively. Of the 107 lipid species identified to be differentially expressed between the three groups, these could be simplified into assaying 98 lipids for the

Chinese, 99 lipids for the Malays, and 97 lipids for the Indians (Supplementary Table 8).

**Clinical, lifestyle, and diet**. Of the 284 clinical, lifestyle, and dietary variables, 199 variables were food item composition from the Food Frequency Questionnaire, with 46 lifestyle indicators on physical activity, and usage of alcohol and tobacco. The analysis of the 39 clinical variables identified 16 variables to be significantly different across the three populations (Fig. 5), with para-umbilical skinfold measurement, body mass index (BMI), and waist circumference emerging as the three most significantly differentiated measurements ($P_{Bonferroni} = 8.80 \times 10^{-13}$, $P_{Bonferroni} = 2.26 \times 10^{-12}$, and $P_{Bonferroni} = 3.37 \times 10^{-12}$, respectively, Supplementary Fig. 7A and Supplementary Table 9). Chinese exhibited the lowest average para-umbilical skinfold measurement (mean = 23.4 mm, SE = 0.50 mm) compared to the Malays (mean = 26.7 mm, SE = 0.53 mm) and Indians (mean = 29.2 mm, SE = 0.50 mm), and this trend was similarly observed in BMI (Supplementary Fig. 7B) and in waist circumference measurement (Supplementary Fig. 7C). Unsurprisingly, there were noticeably high levels of correlation between most of the 16 variables, especially among anthropometric traits as well as variables predictive of metabolic health. This pattern of correlation was also consistent across all three ethnic groups (Supplementary Fig. 8A–C). The proportion of ever-smokers was significantly higher in Malays (34.5%) than to the Chinese (27.9%) and Indians (22.5%).

The first three principal components from the eigen-decomposition of the information from the 199 dietary variables accounted for 10.0% of the variation, even though it will require 115 principal components to explain at least 90% of the variation in the 199 variables. When we inspected the loadings for the first principal component, larger positive loadings were observed in food items common in the diet of Indians (such as dhal, fish/meat curry without coconut and dosai), whereas larger negative loadings were observed in food items common of Chinese diet (such as dim sum, roasted/grilled/BBQ meat, stir-fried dishes with oyster sauce, Supplementary Table 10). The second principal component was positively loaded at food items representative of Chinese cuisine (such as chicken broth, steamed dishes, and soup dishes) and negatively loaded at a mixture of Malay and Indian food items (such as nasi lemak and nasi briyani, Supplementary

**Table 4 Five most differentiated miRNAs between three ethnic groups after adjusted for RT plate effect**

| miRNA | P-value | lsm$_{Chinese}$ | lsm$_{Malay}$ | lsm$_{Indian}$ | FC $_{(Malay-Chinese)}$ | FC$_{(Indian-Chinese)}$ |
|---|---|---|---|---|---|---|
| has_miR_4732_3p | 9.50E-04 | 18.01 | 18.17 | 17.42 | 1.12 | 0.67 |
| hsa_miR_375 | 9.40E-03 | 18.04 | 17.52 | 17.38 | 0.70 | 0.63 |
| hsa_miR_140_3p | 1.10E-02 | 21.82 | 22.00 | 21.40 | 1.13 | 0.75 |
| hsa_miR_378a_3p | 2.92E-02 | 20.81 | 20.87 | 20.48 | 1.04 | 0.79 |
| hsa_miR_378a_5p | 3.13E-02 | 15.10 | 15.35 | 14.69 | 1.19 | 0.75 |

*Note:* Least squares mean (lsm) was calculated for each ethnic groups and fold change was also calculated with respect to Chinese FC is calculate in this way: since the lipid data was log-2-transformed, i.e., $\log_2 FC_{(Malay-Chinese)} = lsm_{Malay} - lsm_{Chinese}$; $FC = 2^{\log 2FC(Malay-Chinese)}$

Table 11). The third axis was positively loaded for a mixture of Malay and Chinese food items (such as deep-fried dishes, dishes in assam pedas/curry with coconut, innards and braised/stewed/roasted dishes) and negatively loaded at Indian food items (such as dhal, Indian bread, and dishes in curry without coconut, Supplementary Table 12).

**Discussion**

This concept paper has introduced the iOmics, which recruited 364 subjects from the three major ethnic groups in Singapore and assayed each of them across a variety of omics technologies which included genomics, transcriptomics (including miRNAs), and lipidomics. Clinical measurements as well as lifestyle data around physical activity and nutrition were similarly available for each of these individuals. In this paper, we have investigated a fundamental hypothesis: to what extent do ethnic differences explain the variation in the expression of the different omic and phenotypic measurements; and subsequently to identify the specific sub-units that segregate between the ethnicities. In many instances, this reduces to the problem of identifying the sub-unit where the expression of a product is more likely to be higher or lower in one ethnic group compared to another, where the product may be a SNP or HLA allele, gene transcriptome activity, lipid species spectrometry measurement, or miRNA count. This is similarly the case in pinpointing the clinical, lifestyle or environmental measurements that are different between the ethnicities. Understanding the extent that populations cluster according to product expression is a foundational assumption that underpins many of the existing databases such as those from the International HapMap Project[4], 1000 Genomes Project[6], and ENCODE[40].

While PCAs were used to illustrate the extent of clustering between samples of different ethnicities, it is important to recognize there were overwhelmingly more datapoints available in the genetic data than in the rest of the omics and non-omics data. A reduced subset of 101,099 SNPs was used in the genetic analysis, which was already more than the 21,649 gene transcripts, 282 lipid species, and 284 non-omic measurements on clinical phenotypes, lifestyle, and diet. What is surprising, however, is that the use of 199 dietary variables alone was able to elucidate clearer patterns of ethnic membership which the larger data sets of transcriptomics and lipidomics were unable to. This perhaps suggests that downstream biological activities such as gene or lipid expression generally tend to be conserved across populations, except of specific sub-units that may have differed owing to biological adaptation to different environmental (including dietary) exposures. This is perhaps unsurprising, as upstream molecular changes in DNA may not necessarily culminate to impact consequential downstream products such as mRNA transcription, protein translation, and eventually influence catalytic reactions affecting metabolites and lipids.

One finding that is consistent across almost all the omics (except miRNAs) and non-omics comparisons is the greater heterogeneity seen between Indians and non-Indians (particularly Chinese), than between Chinese and Malays. While this concurred with previous reports[7, 33, 41], what the iOmics has shown is that even among just the Indians, there is a lot more biological heterogeneity than within the Chinese or the Malays separately. This was hinted in an earlier article looking at the genetic diversity exhibited by whole-genome sequencing a subset of the Indians in the iOmics[31], where when adjusted for the sample size, the same number of South Asian Indians was considerably more heterogeneous genetically compared to the same number of Southeast Asian Malays or Han Chinese. The iOmics confirmed that the greater intra-population diversity exhibited by the Indians is not simply confined to genetics, but is similarly seen in the lipidomics profiles, as we observed the correlations between lipid species were weaker in the Indians than in the Chinese and Malays.

Data resources in the life sciences have been instrumental in driving the progress biomedical and clinical research. Such data infrastructure can often be benchmarked against three kinds of impact that they deliver[42, 43]: (i) scientific impact—information from such databases typically provides foundational knowledge that aids the design of future experiments[44–47]; (ii) translational impact—information from such databases typically guides the changing of practices in clinical medicine, highlighting clinical validity and industry relevance[48–50]; and (iii) implementation impact—information from such databases offer insights that guide the development and evaluation of healthcare institutional or governmental policies, by enabling and/or facilitating health services and health systems research, especially those pertaining to financing and regulatory approvals[51]. Notably, the impact that a data resource delivers is not necessarily exclusive to a single category, and well-designed and curated databases can often deliver impact spanning all three categories. It is with this in mind that the iOmics was designed, particularly with the ability to recall subjects according to their omics and non-omics profiles, allowing the further expansion of the iOmics database whenever newer technologies have been proven to deliver information of value.

This paper has only scratched the surface on what the iOmics resource can deliver. Evidently there is considerable potential in the use of this data set to investigate the degree of co-expression that exists between the different omics measurements. The relatively small sample size will undoubtedly hinder the discovery of networks with modest levels of co-expression. In addition, there is a dire need for novel methodologies to be designed with the specific intent of addressing the problem of multiple tests, which invariably is present in such cross-omics analyses. But like the HapMap before this, identifying clear patterns of co-expression that are ubiquitously present across all three ethnic groups is a real possibility. The aspiration for the iOmics will be to integrate the present resource with longitudinal and prospective clinical records, where clinical decisions can be made not only to address clinical needs, but also with reference to the baseline omics, lifestyle, and nutritional profiles.

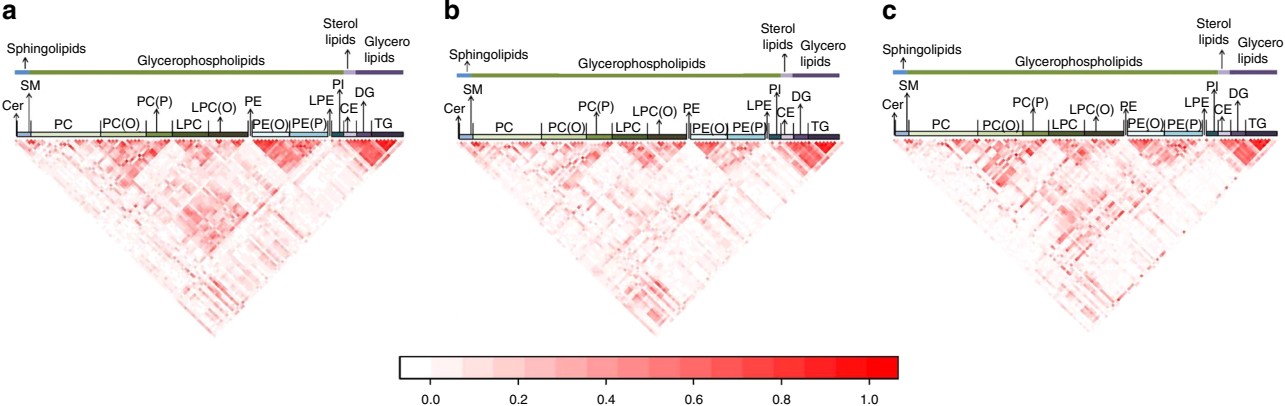

**Fig. 4** Correlation heatmap of the 282 lipids in each ethnic group. The correlation heatmap between 282 lipids in the **a** Chinese; **b** Malays; and **c** Indians. The correlation was calculated by using concentration of the lipids via Pearson's correlation, $r^2$. The lipids are first categorized into lipid category and followed by lipid classes. In each of the lipid class, the lipid species are ordered according to their carbon chain length and degree of unsaturation (number of double bonds). The intensity of the color reflects the magnitude of the correlation, in which the *white* color means $r^2 = 0$ and *red* color means $r^2 = 1$

## Methods

**Samples**. Subjects enrolled in the iOmics were originally recruited for a community-based multi-ethnic prospective cohort that is part of the SPHS project (formerly Singapore Consortium of Cohort Studies), where 122 Chinese, 120 Malays, and 122 Indians were randomly selected. Subjects in SPHS were recruited to participate in the National Health Survey, which involved a random age-stratified and gender-stratified sampling of Singapore residents living across the country in order to generate a representative sample to understand the health status of Singapore residents in the country. All subjects were between 40 and 65 years old at the point of recruitment in 2008, and did not possess any pre-existing major health conditions, defined in this study to include cardiovascular disease, mental illness, diabetes, stroke, renal failure, hypertension, and cancer. However, no detailed clinical assessments were performed to confirm the absence of above-stated diseases, and ascertainment depended on self-reports. The ethnic membership of each subject was assigned after verbal confirmation that all four grandparents belonged to the same ethnicity. Blood sampling for the lipidomics and transcriptomics assays was performed after at least 12 h of fasting. All study subjects provided written informed consent for the participation, and all protocols associated with this study were approved by the National University of Singapore Institutional Review Board.

**Clinical, lifestyle, and diet data**. All participants were required to complete a health survey, a health screening, and a food frequency questionnaire, all of which were administered by trained interviewers and nurses. Clinical measurements such as weight, height, fasting glucose, glycated hemoglobin (HbA1c), blood pressure, and lipids were recorded, and blood and urine samples were taken after at least 12 h of fasting. The food frequency questionnaire comprises a 199-question survey that was validated for use in Singapore across the three ethnic groups[52], while the health survey covered a total of 79 questions on tobacco use, health status, and physical activity. A total of 335 variables were measured across clinical, lifestyle, and diet, although we excluded variables with > 20% missing entries across all the iOmics individuals, and we excluded individuals who possessed > 20% non-valid or missing entries across the remaining variables. The final data set for clinical lifestyle and diet comprised 122 Chinese, 116 Malays, and 120 Indians measured across 284 variables, of which there were (i) 39 clinical variables; (ii) 46 variables related to lifestyle; and (iii) 199 variables related to diet. The data for these 284 variables across all 358 individuals is publicly available at //phg.nus.edu.sg/#iomics.

**Genetics—genome-wide SNP genotyping**. Genomic DNA of 350 individuals (111 Chinese, 120 Malays, 119 Indians) and 348 individuals (111 Chinese, 119 Malays, 118 Indians) were assayed on the Illumina Omni 2.5 and Illumina Exome microarrays, respectively. QC of both sets of genetic data were performed in the following four phases in sequential order: (1) SNPs from both arrays were combined to yield a single data set for every individual, where for overlapping SNPs, the genotypes from the microarray with the least amount of missingness were retained; (2) sample duplicates, related samples, or samples with missingness > 2% were removed; (3) samples with inconsistent population membership between the self-reported ethnicity and genetically inferred ethnicity were removed; (4) SNPs with high degree of missingness (> 5%) and gross departure from Hardy–Weinberg equilibrium within each ethnic group ($P_{HWE} < 10^{-3}$) were removed. This produced a final set of 2,527,458 unique SNPs (2,299,708 from Omni 2.5, 227,750 from the Exome chip) across 110 Chinese, 108 Malays, and 105 Indians.

**Genetics—pharmacogenomics SNP genotyping**. In addition to genome-wide genotyping using the commercial Illumina microarrays, a customized Infinium genotyping assay (Illumina, San Diego, CA, USA) was also designed to probe 4534 SNPs in 350 selected genes involved in drug absorption, distribution, and excretion[33]. Genotypes were called using the proprietary Illumina Genome Studio software package. QC procedures included removing duplicate samples with the lower call rate, or samples with less than 90% of the SNPs successfully called. SNPs were excluded if the call rates were less than 90%, or if concordance was less than 95% for SNPs that were also found on the post-QCed data for Omni 2.5 or the Exome chip. The post-QC data comprised 4032 pharmacogenomic variants across 106 Chinese, 112 Malays, and 115 Indians.

**Genetics—HLA classical alleles typing**. A high-resolution sequence-based HLA typing was performed on the three Class I loci (-A, -B, -C) and five Class II loci (-DPA1, -DPB1, -DQA1, -DQB1, -DRB1) with a target resolution of at least four digits using a sequence-based typing method with taxonomy-based sequence analysis[53, 54]. A total of 198 HLA alleles were observed across 111 Chinese, 119 Malays, and 120 Indians.

**Lipidomics**. The plasma samples preparation and lipid extraction were followed according to what were previously described[55]. Lipid sample of all individuals was injected into an Agilent 1200 LC system with combined of an Agilent 6490 triple quadrupole (QQQ) instrument (Agilent Technologies, Santa Clara, CA) for liquid chromatography electrospray ionization-tandem mass spectrometry method. The lipid species measured in this study can be found in Supplementary Data 1. The liquid chromatography was performed on 1 µl of lipid extract using a Agilent Zorbax C18, 1.8 µm, 50 × 2.1 mm column at 400 µL/min using the following gradient condition: (i) 0% B to 40% B over 2 min, then 100% B over the next 6.5 min, (ii) 0.5 min at 100% B, (iii) a return to 0% B over 0.5 min then 0.5 min at 0% B prior to the next injection. Both solvent A and B consisted of 10 mM NH4COOH with tetrahydrofuran: methanol:water in the ratio of (i) 20:20:60 and (ii) 75:20:5. Subsequently, precursor ion scans and neutral loss scans were conducted as to identify the lipid species present in human plasma. Next, multiple-reaction monitoring in positive mode[56] was conducted to quantify lipid species (Supplementary Data 1). The concentrations of the lipid were then calculated by relating the peak area of each lipid species to the peak area of the corresponding internal standard. The Phosphatidylinositols (PI), alkenylphosphatidylethanolamines (PE(P)), cholesterol esters (CE), DG, and TG species were corrected for response factors that determined for each species[57]. The nomenclature was used for lipid species, for instance, a lysophosphatidylcholine with a Fatty Acid that contains 22 carbons and 6 double bonds as 22:6, which followed the LIPID MAPS nomenclature[58] and recent revisions by Liebisch et al[59]. The mass spectrometry data was acquired on Agilent Mass Hunter Acquisition software and extraction of the lipid data was processed using Agilent Mass Hunter QQQ Qualitative and Quantitative Analysis software vB.07.000 (Agilent Technologies Corp., Santa Clara, CA). The QC was performed in two steps: (1) lipid species with signal-to-noise ratio of greater than three, compared to lipid species signals in blank samples were retained; (2) common lipid species with coefficient of variation percentage of less than 25%, which were widely accepted standard were retained. No instrumental drift was observed during the course of the MS analytical run (data not shown). In this study, lipid concentrations were reported as relative concentrations, the detailed description has previously been described[56]. It is an indication of the relative abundance of each lipid species or class[56]. The relative concentrations of lipid classes and subclasses were subsequently calculated from the sum of individual lipid

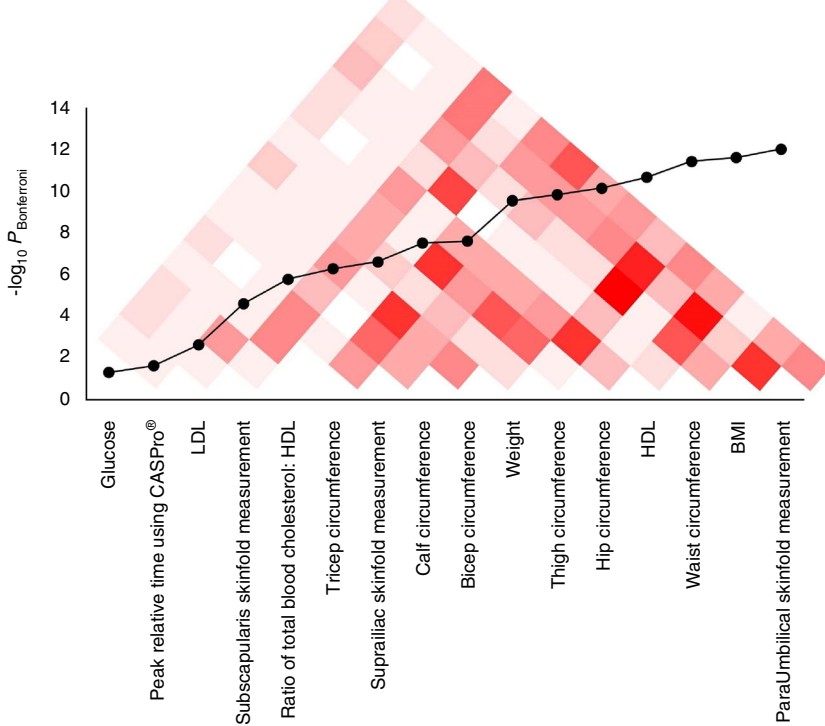

**Fig. 5** Distribution of the differences and correlation of the 16 significant differentiated clinical phenotypes across three populations. The distribution of the differences of the 16 clinical significant phenotypes across three populations, with a correlation heatmap of the phenotypes. The correlation was calculated using Pearson's correlation, $r^2$, across 358 samples. The intensity of the color reflects the magnitude of the correlation, in which the *white* color means $r^2 = 0$ and *red* color means $r^2 = 1$. The details of the differences can be found in Supplementary Table 9

species within each class[57]. A total of 282 lipid species were measured in 122 Chinese, 117 Malays, and 120 Indians, where the lipid data in pmol per ml plasma were log2-transformed for downstream analyses.

**Transcriptomics—mRNA transcripts experimental analysis.** Whole blood of all 364 individuals was used for RNA isolation using the Tempus 12-Port RNA Isolation Kit (Applied Biosystems/Ambion, Carlsbad, CA), according to the manufacturer's instructions. Total RNA yield was quantified using a Nanodrop ND 1000 spectrophotometer (Nanodrop Technologies, Wilmington, DE), and RNA integrity number was measured with the Agilent 2100 Bioanalyzer using RNA 6000 Nano chips (Agilent Technologies Inc., Santa Clara, CA). cDNA was synthesized and amplified from 200 ng RNA using the Applause WT-Amp System (NuGEN Technologies Inc., San Carlos, CA) and hybridized to Affymetrix Human Gene 1.0 ST arrays (Affymetrix Inc., Santa Clara, CA). All sample labeling, hybridization, and image scanning were performed according to the manufacturer's instructions. The quality of the gene expression data was assessed in the following two phases in sequential order: (1) probesets QC to remove non-autosomal probes and to identify a set of unique probes that were expressed in at least one sample; (2) sample QC to remove low-quality samples, outliers, and ambiguous samples (see Supplementary Methods for details). This produced the post-QC data set of 21,649 probesets at 98 Chinese, 75 Malays, and 96 Indians.

**Non-coding RNAs (miRNAs) profiling.** miRNA biomarker profiling was performed with a patented mSMRT-qPCR miRNA assay (MIRXES) in a highly controlled workflow. The miRNAs profiling process was performed in the following phases: (i) total serum RNA (up to 200 μl) was extracted using the miRNeasy serum/plasma miRNA Isolation kit (Qiagen, Hilden, Germany) on a semi-automated QiaCube system; (ii) a set of three proprietary spike-in control RNAs (~20 nt, MIRXES) with sequences distinct from annotated mature human miRNAs (miRbase version21) was added into the sample lysis buffer prior to RNA isolation; (iii) the quantified levels of the spike-in control RNAs were used to normalize RNA isolation efficiency; (iv) the isolated miRNAs were then reverse transcribed using miRNA-specific RT primers per manufacturer's instruction (MiRXES); (v) a 6-log serial dilution of synthetic templates for each miRNA and a non-template control were concurrently reverse transcribed; (vi) sample and template cDNAs were then pre-amplified through a 14-cycle PCR reaction using Augmentation Primer Pools (MiRXES). In each amplified cDNA sample, a total of 300 candidate miRNAs were measured by qPCR using miRNA-specific qPCR assays (MIRXES), with technical replicates on ViiA7-384-well qPCR system (Applied Biosystems). Upon the completion of profiling, raw threshold cycle (Ct)

values were determined using the ViiA™ 7 RUO software with automatic baseline setting and a threshold of 0.5 and absolute copy numbers of each miRNA were determined through interpolation of the Ct values to that of the synthetic miRNA standard curves and adjusted for RT-qPCR efficiency. Technical variations introduced during RNA isolation and the process of RT-qPCR were normalized using the spike-in control RNAs. We excluded any miRNAs with ≤90% call rate across all 364 samples, resulting in a final panel of 274 miRNA variants across 117 Chinese, 115 Malays, and 119 Indians for downstream analyses. All subsequent analyses were performed on normalized (via global mean normalization) and log-2 transformed miRNA expression values.

**Principal component analysis.** A series of PCAs were performed using the different sources of omic data as well as with a combination of clinical, lifestyle, and diet data to identify the presence of cryptic relatedness between the subjects. The PCA with the genetic data was performed with smartPCA and EIGENSOFT[60] using a subset of 101,099 pseudo-randomly chosen SNPs selected across the 22 autosomal chromosomes to minimize linkage disequilibrium between the SNPs. The remaining PCAs were performed using eigen-decomposition of the respective $N \times K$ matrices, where: (i) for lipids, $N = 359$ samples and $K = 282$ lipid species; (ii) for transcriptomics, $N = 269$ samples and $K = 21,649$ transcript probesets; (iii) for miRNA, $N = 351$ samples and $K = 274$ miRNAs; (iv) for the combination of clinical, lifestyle, and diet, $N = 358$ samples and $K = 284$ variables. A separate PCA was also performed for the 358 samples using only the 199 dietary variables to evaluate the extent that the individuals cluster according to their dietary responses.

**Identifying inter-ethnic variation with analysis of variance (ANOVA).** An ANOVA was used to identify the sub-units within each omic technology that segregated across the three ethnic groups, testing the null hypothesis that the mean levels of the sub-unit were exactly identical across all three ethnic groups, against the alternative hypothesis that at least one of the three ethnic groups exhibited a different mean. Gender was adjusted in the analyses with the lipid, transcription, and clinical/lifestyle/diet data. Owing to the different number of tests considered in each technology, we declared statistical significance if the within-omic Bonferroni-corrected P-value was less than 0.05.

**Identifying inter-ethnic variation with genetic data.** The Wright's $F_{ST}$[61, 62] was used to quantify the extent of allele frequency differences between the ethnic

groups at each genetic variant, as measured by

$$F_{ST} = \frac{(k-1).\sigma^2}{k.\overline{p}.(1-\overline{p})},$$

where $\sigma^2$ denote the variance of the frequency of a particular SNP or HLA allele across the three populations, $\overline{p}$ denote the mean frequency of the same allele in the three populations, and $k$ denote the total number of populations. Here, each of the HLA classical alleles at every Class I (HLA-A, -B, -C) and Class II (-DPA1, -DPB1, -DQA1, -DQB1, and –DRB1) loci was considered as a distinct allele from a biallele SNP in order to calculate the $F_{ST}$ value for that HLA allele. The $F_{ST}$ values of the 2,527,458 SNPs from the Omni 2.5 and Exome microarrays were used to derive a genome-wide distribution, and the $F_{ST}$ values of the 4032 pharmacogenomics SNPs were mapped against the genome-wide distribution to derive the empirical $P$-value, calculated by

$$P_{empirical} = \frac{\text{Number of genome} - \text{wide SNP with } F_{ST} > \text{Observed } F_{ST}}{\text{Total number of SNPs}}.$$

In order to identify contiguous stretches of the genome that are most differentiated across the three populations, we derived a region-based statistic based on the degree of over-representation of high $F_{ST}$ SNPs in a pre-defined genomic window (100 kb non-overlapping window), and quantified the degree of over-representation with a Binomial probability[63]. For the 4032 pharmacogenomic SNPs, we identify SNPs that exhibited empirical $P$-values < $10^{-4}$, where the threshold is conservatively chosen to account for both multiple testing and linkage disequilibrium between the SNPs.

**Mapping correlation patterns in lipidomics profiling**. To assess the pattern of correlation in the lipidomics expression, we calculated the Pearson correlation coefficient ($r$) for the lipid profiles of every pair of the 282 lipid molecules. This analysis of correlation pattern was performed separately in each of the three populations, and we considered a lipid species to be 'tagged' in a population if it exhibited a squared correlation coefficient $r^2 > 0.80$ as calculated by the software CLUSTAG[64].

**Data availability**. The entire set of post-QCed iOmics data is available publicly for download at //phg.nus.edu.sg/#iomics, and the raw data for the different omic platforms are also available upon request to statyy@nus.edu.sg. The genotype data, gene expression data, lipid data, and miRNA data have been deposited at the European Genome-phenome Archive (//www.ebi.ac.uk/ega/), which is hosted by the EBI, under accession number EGAS00001002527.

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

## Acknowledgements

This project acknowledges the support of the Saw Swee Hock School of Public Health, Yong Loo Lin School of Medicine, National University Health System, and the Life Sciences Institute under the Office of Deputy President for Research and Technology, National University of Singapore. W.Y.S. and Y.Y.T. additionally acknowledge support from the Biomedical Research Council (grant 03/1/27/18/216), National Medical Research Council (grant 0838/2004), National Research Foundation (through the Biomedical Research Council, grants 05/1/21/19/425 and 11/1/21/19/678).

## Author contributions

Y.Y.T. conceived and designed the study. W.Y.S. and Y.Y.T. wrote the manuscript. W.Y.S. performed data analysis with contribution from E.T., H.B., L.Z., R.Z., C.H., S.L.C., L.P.W., and N.E.P., W.X., D.K.N.M., Y.L., B.L., L.W.L.T., T.A.P., T.B., P.J.M., P.A.M., W.Y.L., M.L., K.S.C., R.T.H.O., L.R.B., C.C.K., H.P.T., R.S., M.R.W., P.L., and Y.Y.T. contributed to population samples, genotyping data, pharmacogenomics SNP genotyping data, HLA classical alleles typing data, lipidomic data, transcriptomic data, miRNAs data, and clinical, lifestyle, and dietary data.

## Additional information

**Competing interests:** The authors declare no competing financial interests.

**Reprints and permission** information is available online at //npg.nature.com/reprintsandpermissions/

