## [Peer Review file · Nature Communications]

Reviewers' comments:

Reviewer #1 (Remarks to the Author):

The manuscript by Saw et al reports a comprehensive multi-omics approach to discover molecular differences between three major ethnic groups of Singapore population. This is an interesting and timely effort that so far received surprisingly little attention of the omics community. I am supportive of the publication of this manuscript, however my major concern is the poor clarity of presentation of these otherwise very interesting and novel data and their descriptive interpretation. Lipidomics data are presented such that they do not allow direct comparison with published lipidomics profiles e.g by LIPIDMAPS (Quehenberger et al , 2010). Were these datasets at all concordant?! Also, it is difficult to grasp if inter-ethnic differences were among major (most abundant) species, or only about statistically significant variations of relatively low abundant species? What are the most specific fold changes – is it in the range of 10x or 10%? Are there any major trends e.g. lipid levels generally higher in one of the ethnic groups? If yes, was anything common among the affected species – saturation of fatty acid moieties? Number of carbon atoms? Lipid class (does not seem to be the case)? What lipid classes remained unaffected (=conserved?). The data are there, but they are scattered throughout the manuscript which makes it very difficult to understand what is, actually, going on? Here Suppl Tables 7 and 8 do not really help.

Minor critiques:

Line 56: pls specify the origin of biomaterial for lipidomic measurements (plasma, biopsies etc)

Line 62: please rephrase, it reads unclear considering there were 3 populations studied

Line 90: please rephrase, lipids per se are not expressed

Line 122: please indicate gender composition for study subjects within each ethnic group. Also, it would be helpful to give (some) reference to BMI as a naïve, yet useful measure of lipid metabolism status

Line 183: sufficient to name it Lipidomics

Lines 184/185: the division in two sites is confusing, particularly because “lipid data from site 1 were used in this study”. As a reader, I would not care where data were made for as long as they are reasonably consistent.

Lines 199 / 200 This will only work if CE is adjusted to m/z of fragmented lipids, otherwise molecules will have different response even within the same lipid class

Line 215 “lipid concentrations were reported as relative concentrations” – there is no such thing as relative concentration. Please provide a clear description of how lipid concentrations were calculated and in what units (%? If yes, normalized to what?) were they expressed? Ref 38 is not helpful.

Line 219 “the spectrometry readouts” – concentrations?! Mol%?! Pls sort it out and explain clearly – this is a key dataset for the entire article

Line 436 “Our analyses also suggested that lipid species with similar mass-to-charge ratio, m/z have a tendency to be co-expressed.” – lipids are not expressed and, furthermore, lipids of similar m/z that belong to different classes have nothing in common.

Line 439 “We identified 29 lipid species in the Chinese that can effectively represent the information found in 71 species” – sorry, I failed to understand it. Species as such represent no information

Line 442 and also Suppl. Table 7: DG and TG are !NOT! sterol lipids

Line 531 and the entire paragraph – this is a bold political statement that does not need to be in a scientific article

Reviewer #2 (Remarks to the Author):

Summary:

The work by Saw et al. is well conducted and well presented. Saw et al. seek to establish a Singaporean molecular population reference database, using genomics, transcriptomics and lipidomics techniques and questionnaires. Their study comprises the three major populations in Singapore; the Chinese, the Malays and the Indians. Previous reports have depicted the genetic structure of e.g. the Netherlands (Francioli et al. Nature Genetics 2014), the UK (Leslie et al. Nature 2015) and Denmark (Besenbacher et al. Nature Communications 2015). Saw et al. conceptually extend their approaches by reporting various levels of biological molecules that differentiate subpopulations. For each platform, they report specific molecular and environmental markers that differentiate any of the ethnic groups.

Major comments:

It is not obvious exactly how a national genome variation project is instrumental in enabling genetics as forerunning in precision medicine. It would strengthen the manuscript if the authors could provide some specific examples on top of the references the listed.

In order to know how general the results from the study are, it would be useful to know how the individuals were recruited in the original cohort study. In other words, are there any biases in the original study which should be reflected on in this manuscript?

Since the authors present their work as a "concept paper" it would be instructive to highlight why these selected "omics" platforms are providing a proper basis to compare ethnic subgroups.

Large-scale RNA-sequencing to cover a higher fraction of all GENCODE genes as well as splice isoforms, and large-scale epigenetics measurements are notably missing and could potentially have identified more distinct regulatory differences between the ethnic groups.

Minor comments:

Methods: Where the blood samples for the lipidomics and transcriptomics taken after fasting and were they taken at the same time of the day for all individuals?

Discussion: miRNA is not an omics technology. Consider rewording the first sentence.

Discussion: References missing in the discussion of the impact of this type of data infrastructure.

Reviewer #3 (Remarks to the Author):

Saw et al. have emphasized the importance of multi-omics datasets that are accompanied by a series of non-omic variables (a total of 284 in the present study) related to the clinical, lifestyle and dietary profiles. The authors have performed a large number of analyses with particular focus on the ethnic variation between three principal ethnic groups in Singapore. Their analyses appear to be comprehensive and the results are of some interest, in particular, from the viewpoint of population genetics (or genomics).

While the authors demonstrated the results for each measurement separately, they did not show

the usefulness of integrating multi-omics datasets, mainly because of an insufficient sample size in the present study alone, as mentioned in the manuscript.

Even if the authors regard this as a concept paper, it is preferable (and more attractive) to the readers to show some examples for co-expression that exists between the different omics measurements and/or non-omic variables. For instance, they can pursue a mechanism underlying the observed transcriptomic difference at urotensin II between Indians and non-Indians, starting from the current set of 364 individuals to expanded, larger samples, as an example to utilize the omics towards integration.

All three reviewers have provided helpful and constructive comments, which we have considered in revising the manuscript. Specifically, we have listed a detailed point-by-point response to explain the action taken in light of the reviewers' comments.

Reviewer 1:

The manuscript by Saw et al reports a comprehensive multi-omics approach to discover molecular differences between three major ethnic groups of Singapore population. This is an interesting and timely effort that so far received surprisingly little attention of the omics community. I am supportive of the publication of this manuscript, however my major concern is the poor clarity of presentation of these otherwise very interesting and novel data and their descriptive interpretation. Lipidomics data are presented such that they do not allow direct comparison with published lipidomics profiles e.g by LIPIDMAPS (Quehenberger et al , 2010). Were these datasets at all concordant?! Also, it is difficult to grasp if inter-ethnic differences were among major (most abundant) species, or only about statistically significant variations of relatively low abundant species? What are the most specific fold changes – is it in the range of 10x or 10%? Are there any major trends e.g. lipid levels generally higher in one of the ethnic groups? If yes, was anything common among the affected species – saturation of fatty acid moieties? Number of carbon atoms? Lipid class (does not seem to be the case)? What lipid classes remained unaffected (=conserved?). The data are there, but they are scattered throughout the manuscript which makes it very difficult to understand what is, actually, going on? Here Suppl Tables 7 and 8 do not really help.

- i) The reviewer commented that the presentation of the lipidomics should be significantly improved, especially to allow for direct comparison with published lipidomics profiles such as that by LIPIDMAPS. In addition, greater clarity is needed to explain whether inter-ethnic differences were observed among major species, or whether in relatively low abundant species. Further clarifications should also be provided on the magnitude of the fold changes, and whether there are any specific lipid species that are systematically different in at least one ethnic group. The report should also focus on lipid classes that are conserved across ethnic groups, in addition to those that are different. The reviewer also commented on the superfluous nature of Supplementary Tables 7 and 8.

Our response:

We thank the reviewer for the comments and apologise for the poor organization of the lipidomics results. We have undertaken a substantial rewrite of the paragraphs in the Results section reporting on the lipidomics findings, and hopefully have now provided a clearer exposition of our findings from the lipidomics dataset. In particular, our previous manuscripts have reported the lipidomics data in pico mol per ml (pmol/ml) plasma, whereas the article by LIPIDMAPS has reported their data in nano mol per ml (nmol/ml) plasma. We have thus converted the units of the results of the lipidomics analyses in our revised manuscript to nmol/ml to make it comparable to current published articles.

The specific paragraphs for the lipidomics analyses in the Results section now read as the following: “The set of 282 lipid species came from four lipid categories (glycerophospholipids, sphingolipids, sterols, neutral lipids), of which there were 20 lipid classes (**Supplementary Table 7**). We identified 107 lipid species where the lipid levels were differentiated between the three ethnic groups, of which 20 lipid species exhibited at least 1.5 fold difference in lipid levels with respect to the Chinese (**Supplementary Table 8** and **Supplementary Figure 5**). Of the 107 differentiated lipid species, 91 were considered low abundant while 16 were major abundant species with average lipid concentration exceeding 10 nmol/ml (**Supplementary Table 8**). The majority of the differentiated lipids (86 lipids) belonged to the glycerophospholipid category which comprised the main component of the biological membrane, serving to (i) stabilize and

isolate the intracellular environment from the external environment; (ii) regulate the transportation of the molecules through the membrane; and (iii) organise membrane component into localised areas involved in specific processes such as signal transduction⁵⁴. Of the 107 differentiated lipid species, 88 were driven by lipid level differences between Indians and non-Indians, and the remaining 19 were driven by differences between Chinese and Malays. In fact, the three most differentiated lipid species were due to differences seen between Indians and non-Indians (**Supplementary Figures 6A-C**), at (i) PC(O-40:7) ($P_{\text{Bonferroni}} = 5.78 \times 10^{-24}$); (ii) PC 38:3 ($P_{\text{Bonferroni}} = 3.84 \times 10^{-21}$); and (iii) PE(O-40:7) ($P_{\text{Bonferroni}} = 3.54 \times 10^{-20}$).

Our analyses also suggested that lipid species with similar chemical properties that were categorized in the same lipid class have a tendency to be correlated. As a result, we investigated the extent of correlation between lipid molecules within each ethnic group, both within lipid class and between lipid classes in each of the three ethnic groups (**Figure 4**). We identified 29 lipid species in the Chinese that can effectively represent the information from the measurements of 71 lipid species (defined as $r^2 > 0.8$). For the Malays, there were 26 tagging lipids for 61 species; and 26 tagging lipids for 60 species for the Indians (see **Supplementary Tables 9-11**). Notably, lipid classes in the sterol (free cholesterol (COH), cholesteryl ester (CE)) and neutral lipids (diglycerides (DG) and triglycerides (TG)) categories were highly correlated within and between classes, with lipids from DG and TG accounting for 22% (28/125), 16% (21/125) and 3% (2/65) of the observed lipid pairings. This meant that the 282 lipid species can be summarised by 240, 247, and 248 ‘tagging’ lipids in the Chinese, Malays and Indians respectively. Of the 107 lipid species identified to be differentially expressed between the three groups, these could be simplified into assaying 98 lipids for the Chinese, 99 lipids for the Malays and 97 lipids for the Indians (**Supplementary Table 12**).”

In response to the reviewer’s comment about how we should also report on lipid classes that remained unaffected across the ethnic groups, we agree with the Reviewer that findings of conserved lipid quantification across ethnicity can be as interesting as findings of differential quantification. However, our present study design does not confer the confidence to report on conservation of lipid quantification across the different ethnic populations, due to the relatively small sample size of our study. We are thus unable to rule out whether the lack of statistically significant differences in inter-ethnic lipid quantification is due to insufficient statistical power or due to genuine underlying biological reasons. As such we have adopted an approach that focuses on reporting observed differences that are statistically significant (after the appropriate adjustment for multiple comparisons), and this means we are more confident on the authenticity of the reported differences (high specificity) but we may not have identified all the underlying biological differences between ethnic populations (lower sensitivity).

As recommended by the reviewer, we have revised Supplementary Tables 7 and 8 in our revised manuscript. The revised Supplementary Table 7 shows the distribution of the lipid species which are categorized in the different lipid classes, which we believe this, will be informative to readers who may not be entirely familiar with lipidomics. The revised Supplementary Table 8 reports the 107 lipid species that are found to be significantly differentiated across the three ethnic groups, after adjusting for gender and where statistical significance accounted for multiple testing with a Bonferroni criterion. We additionally reported the average lipid levels in log₂ pmol/ml scale for each ethnic group, and reported the fold changes observed in the Malays and Indians with respect to the lipid levels in the Chinese. In addition, we have represented the information in Supplementary Table 8 as a figure in Supplementary Figure 5, which illustrates the $-\log_{10} P_{\text{Bonferroni}}$ against the log₂(Fold Change) for the 107 lipid species. The figure and corresponding figure legend are included below:

Supplementary 5. Distribution of the 107 significant lipid species with their respective fold-changes (FCs)

Fold change was calculated with respect to Chinese, and the vertical lines at ± 0.58 correspond to 1.5 fold difference when compared against the Chinese. The circles indicate the FCs of the levels of lipid species between the Malays and Chinese, while the triangles correspond to the comparison between the Indians and Chinese. Numerical quantification of the information in this figure can be found in **Supplementary Table 8**.

We will also like to clarify that there is a separate manuscript that has performed a deeper analysis of the lipidomics data, which we have shared a working draft. Our current manuscript is intended as a concept manuscript to introduce the Singapore Integrative Omics Study and to provide preliminary insights into the depth and capabilities of the different omics datasets. It is beyond the scope of a single manuscript to provide the necessary depth of analyses into the inter-ethnic differences for each separate omics technology. Indeed this was the approach taken for the sequencing, HLA data and early lipidomics datasets, where separate manuscripts have provided timely in-depth reporting of the findings and the datasets have been made available for public sharing:

- Wong, L.P. *et al.* (2013) Deep whole-genome sequencing of 100 Southeast Asian Malays. *American Journal of Human Genetics* 92:52-66.
- Wong, L.P. *et al.* (2014) Insights into the genetic structure and diversity of 38 South Asian Indians from deep whole-genome sequencing. *PLoS Genetics* 10:e1004377.
- Pillai, N.E. *et al.* (2014) Predicting HLA alleles from high-resolution SNP data in three Southeast Asian populations. *Human Molecular Genetics* [epub 15 April]
- Brunham, L. *et al.* (2014) Pharmacogenomic diversity in Singaporean populations and Europeans. *The Pharmacogenomics Journal* [epub 27 May]

- Tantoso, E. *et al.* (2014) Evaluating the coverage and potential of imputing the exome microarray with next-generation imputation using the 1000 Genomes Project. *PLoS ONE* 9:e106681.
- Sapari, N.S. *et al.* (2014) Feasibility of low-throughput next generation sequencing for germline DNA screening. *Clinical Chemistry* [epub 22 October]
- Kapoor, R. *et al.* (2015) Reducing hypersensitivity reactions with HLA-B*5701 genotyping before abacavir prescription: clinically useful but is it cost-effective in Singapore? *Pharmacogenetics and Genomics* 25:60-72.
- Begum, H. *et al.* (2016) Discovering and validating between-subject variations in plasma lipids in healthy subjects. *Scientific Reports* 6:19139.

We can assure the reviewer that the lipidomics analyses were not made in a superficial manner, and a detailed exposition of the lipid differences between the ethnic populations will be made in the separate manuscript which we have attached a working draft as an accompanying document for review. However, we believe it will be beyond the scope of the current concept manuscript to provide a report that will do justice to the scale of data and depth of analyses for each omics technology.

Line 56: pls specify the origin of biomaterial for lipidomic measurements (plasma, biopsies etc)

- ii) The reviewer commented that the manuscript should state the origin of biomaterial for lipidomic measurements (whether plasma or biopsies).

Our response:

We thank the reviewer for the recommendation and we apologise that the nature of biomaterial sampling was not made clear in our earlier manuscript. We will like to clarify that the biomaterial used for lipidomics assaying is plasma, and we have made sure that this is properly documented in the Materials and Methods section of the revised manuscript under the sub-heading of Lipidomics. We have further described the nature of blood sample collection and plasma processing in the Supplementary Material for reproducibility.

Line 122: please indicate gender composition for study subjects within each ethnic group. Also, it would be helpful to give (some) reference to BMI as a naïve, yet useful measure of lipid metabolism status

- iii) The reviewer suggested that the gender composition for the study subjects in each ethnicity should be reported, and that some referencing should be made to BMI as a naïve yet useful measure of lipid metabolism status.

Our response:

We thank the reviewer for the helpful suggestions. We have actually reported the gender composition for the study subjects in each ethnicity for each of the different technology in Supplementary Table 3. We agree with the reviewer that this information is important, as it is entirely plausible that gender differences can confound the nature of expression for certain omics measurement. Similarly, the BMI can be a possible confounder, which is why we have reported the mean (and corresponding standard error) of the BMI for each ethnic group in Supplementary Table 13.

Line 62: please rephrase, as it reads unclear considering there were 3 populations studied

Line 90: please rephrase, lipids per se are not expressed

Line 183: sufficient to name it Lipidomics

Lines 184/185: the division in two sites is confusing, particularly because "lipid data from site 1 were used in this study". As a reader, I would not care where data were made for as long as they are reasonably consistent.

- iv) The reviewer has made several recommendations on the revision of sentences and phrases in different parts of the manuscript.

Our response:

We thank the reviewer for the careful review of our manuscript and for identifying the parts that are presently ambiguous. We have revised the specific sections highlighted by the reviewer, as well as undertaken a careful review of the manuscript to improve and ensure clarity in the rest of the manuscript.

Lines 199 / 200 This will only work if CE is adjusted to m/z of fragmented lipids, otherwise molecules will have different response even within the same lipid class

- v) The reviewer commented that it is necessary to adjust the CE to m/z of fragmented lipids, as otherwise lipid molecules may exhibit different responses even if they are within the same lipid class.

Our response:

We thank the reviewer for this expert comment. We concur with this comment and we will like to assure the reviewer that the analyses of the lipids data have accounted for the fact that different species of the CE class can exhibit different levels of response. Specifically, the calculation method adopted is identical to that from Meikle, P.J. *et al.* (2015) *Journal of Nutrition* 145:2012-2018, where the first author of the published article, Peter Meikle, is a contributing member to the iOmics study and the adjustments have been made specific to the Agilent 6490 instrument used in the iOmics study.

Line 215 “lipid concentrations were reported as relative concentrations” – there is no such thing as relative concentration. Please provide a clear description of how lipid concentrations were calculated and in what units (%? If yes, normalized to what?) were they expressed? Ref 38 is not helpful.

Line 219 “the spectrometry readouts” – concentrations?! Mol%?! Pls sort it out and explain clearly – this is a key dataset for the entire article

- vi) The reviewer commented that it is incorrect to describe lipid concentrations as relative concentrations, and that a clearer description should be provided on how lipid concentrations were calculated and in what units.

Our response:

We thank the reviewer for the comment that the manuscript should clearly indicate how lipid concentrations were calculated. We have revised the section on lipidomics to indicate that the lipid concentrations were measured in pmol/ml plasma. As we did not have stable isotope internal standards available for each individual lipid species, our lipidomics measurements do not reflect absolute lipid concentrations and are semi-quantitative in nature.

Line 436 “Our analyses also suggested that lipid species with similar mass-to-charge ratio, m/z have a tendency to be co-expressed.” – lipids are not expressed and, furthermore, lipids of similar m/z that belong to different classes have nothing in common.

- vii) The reviewer indicated that it is not appropriate to state that lipids are expressed and that lipids of similar m/z from different classes have nothing in common.

Our response:

We thank the reviewer for this comment and we have corrected the manuscript to avoid mentioning that lipids are expressed. Our intention is to report that lipid species with similar chemical properties and which are categorized in the same lipid class have a tendency for their measurements to be correlated. This was a finding from our lipidomics analyses, thus presenting the opportunity to identify whether a few representative lipid species may be able to efficiently summarise the information of the entire lipid panel. We have however rephrased the sentence to ensure a more appropriate representation, which now reads as:

“Our analyses also suggested that lipid species with similar chemical properties that were categorized in the same lipid class have a tendency to be correlated.”

Line 439 “We identified 29 lipid species in the Chinese that can effectively represent the information found in 71 species” – sorry, I failed to understand it. Species as such represent no information

viii) The reviewer commented that the statement “we identified 29 lipid species in the Chinese that can effectively represent the information found in 71 species” to be ambiguous, as species do not present any information.

Our response:

We apologise for the ambiguous statement. Our intention was to identify a smaller subset of lipid species where their measurements are sufficiently correlated with the measurements of a larger subset of lipid species, such that it will be redundant to measure all the lipid species in the larger subset. If this is consistently true across different population groups, then this presents the opportunity that further lipidomics experiments may not need to measure a larger panel of lipid species, but instead a smaller study can be designed where one only performs the lipidomics measurements at a set of well-chosen lipid species. Our study thus evaluated the lipidomics measurements made across the 282 lipid species in the Chinese, Malays and Indians, and observed that there are lipid species that are highly correlated. In theory, if the measurements made at two lipid species are perfectly correlated in all three ethnic groups, then it presents the possibility that one can reduce the complexity of the lipidomics experiment by measuring only one of the two lipid species. This was what we reported, when we stated that 29 lipid species in the Chinese are sufficient to represent the information seen in the measurements made at 71 lipid species. We have however revised our phrasing to be more scientifically accurate, which now reads as:

“We identified 29 lipid species in the Chinese that can effectively represent the information from the measurements of 71 lipid species.”

We also checked and ensured that the definition and methodology to identify representative lipid species are properly clarified in the Materials and Methods section, where we have a sub-section that reads as:

“Mapping correlation patterns in lipidomics profiling

To assess the pattern of correlation in the lipidomics quantification, we calculated the Pearson correlation coefficient (r) for the lipid profiles of every pair of the 282 lipid molecules. This analysis of correlation pattern was performed separately in each of the three populations, and we considered a lipid species to be ‘tagged’ in a population if it exhibited a squared correlation coefficient $r^2 > 0.80$ as calculated by the software *CLUSTAG*⁴⁶.”

Line 442 and also Suppl. Table 7: DG and TG are !NOT! sterol lipids

ix) The reviewer commented that the classification of DG and TG are incorrect, as they are not sterol lipids.

Our response:

We thank the reviewer for the careful review of the manuscript and we apologise for the error. We have corrected our manuscript to correctly state that DG and TG are neutral lipids. This correction has also been performed for Supplementary Table 7. The statement in the Results section of the main text now reads as:

“Notably, lipid classes in the sterol (free cholesterol (COH), cholesteryl ester (CE) and neutral (diglycerides (DG) and triglycerides (TG)) categories were highly correlated within and between classes, with lipids from DG and TG accounting for 22% (28/125), 16% (21/125) and 3% (2/65) of the observed lipid pairings.”

Line 531 and the entire paragraph – this is a bold political statement that does not need to be in a scientific article

x) The reviewer commented that the paragraph in the Discussion section describing the different impact that reference databases can provide to be a bold political statement, and may not be relevant in a scientific article.

Our response:

We thank the reviewer for pointing out the potential inappropriateness of the paragraph. We respectfully suggest to the reviewer that this paragraph should be retained, especially as we believe this

is an often-underlooked aspect of research data infrastructure. There is increasing recognition that databases that are open-sourced and public, and where the sampling and methods of data generation are properly curated and documented, are vital for progress in science to be made. Importantly, progress in science is not solely restricted to fundamental discoveries but also to translational and implementational sciences. In fact, Reviewer 2 even commented that references should be provided to support some of the statements discussing the impact of such data infrastructure. We also like to highlight that in a recent high-profile Correspondence article in Nature (Nature 2017, 543:179) by The Global Life Sciences Data Resources Working Group, and the accompanying full article in bioRxiv (bioRxiv 110825: Towards coordinated international support of core data resources for the life sciences), this was precisely highlighted that the value and impact of life sciences databases serve a multitude of functions, especially in translation and implementation. As such, we believe our paragraph is not a political statement, but rather a scientifically factual statement that is supported by the examples and references that we have included. As such, we will like to appeal to the reviewer to see the appropriateness of such a paragraph in the Discussion section of this manuscript.

Reviewer 2:

Major comment: It is not obvious exactly how a national genome variation project is instrumental in enabling genetics as forerunning in precision medicine. It would strengthen the manuscript if the authors could provide some specific examples on top of the references the listed.

- i) The reviewer commented that in the beginning paragraph of the Introduction, it is not obvious how exactly a national genome variation project is instrumental in enabling genetics to be a forerunner in precision medicine, and recommended that this assertion can be strengthened if some specific examples can be provided.

Our response:

We thank the reviewer for the recommendation to provide greater depth in how genetics has become the forerunner in precision medicine as a result of the availability of national genome variation projects. We agree that this will be important to set the context for the entire manuscript which describes the baseline resource of multiple omics technologies. We have expanded the opening paragraph of the Introduction, which now reads as:

“Knowledge of the genetic determinants of common human diseases has increased tremendously in the past decade, mostly from discoveries made by genome-wide association studies (GWAS)¹⁻³. The efficient design of GWAS for querying the entire genome benefitted from the arrival of the HapMap resource, which queried a large number of genetic markers in groups of individuals pseudo-randomly sampled from a few countries. The result of this endeavour was the production of a map outlining the correlation patterns in the human genome for identifying tagging single nucleotide polymorphisms (SNPs)^{4,5}. The HapMap resource also provided a public database on how prevalent specific alleles are in different ancestry groups in the world⁶. The ability to compare the extent of linkage disequilibrium (LD) differences between ancestry groups also meant index findings from GWAS can be evaluated for reproducibility across multiple population groups^{7,8}. The subsequent development of national genome variation projects has thus produced numerous public databases that have been instrumental at enabling genetics as a forerunner in precision medicine⁹⁻¹². For instance, the predecessor of iOmics, the Singapore Genome Variation Project⁹ which only focused on making static genetic SNP and HLA measurements, indeed facilitated numerous investigations into the population genetics and genetics of common diseases in Asian communities, while at the same time allowed cost-effectiveness assessments and burden estimation of pharmacogenetic testing prior to initiate drug treatments^{13,14} which consequentially influenced policies on governmental subsidies for the costs of genetic tests¹⁵. ”

Major comment: In order to know how general the results from the study are, it would be useful to know how the individuals were recruited in the original cohort study. In other words, are there any biases in the original study which should be reflected on in this manuscript?

- ii) The reviewer commented that more information should be provided on how the individuals were recruited in the original cohort study and whether there were any biases in the original study, as this will allow a clearer understanding of how generalizable the results from the study will be.

Our response:

We thank the reviewer for this pertinent comment. We agree with the reviewer that it is important to provide a clear exposition of how the sampling in the original cohort study was performed and whether there are any inherent biases that may affect the generalizability of the results of the current integrative omics study. We can clarify that there are no inherent biases associated with the sampling of the subjects, as the participants in the original Singapore Population Health Studies (SPHS) study were recruited as part of a National Health Survey meant to undertake a representative sampling to understand the health status of the country in 2008. We have thus provided a further explanation of how the sampling was performed in the Materials and Methods section of the manuscript which reads: "Subjects enrolled in the iOmics were originally recruited for a community-based multi-ethnic prospective cohort that is part of the Singapore Population Health Studies (SPHS) project (formerly Singapore Consortium of Cohort Studies), where 122 Chinese, 120 Malays and 122 Indians were randomly selected. Subjects in SPHS were recruited to participate in the National Health Survey, which involved a random age- and gender-stratified sampling of Singapore citizens living across the country in order to generate a representative sample to understand the health status of Singapore residents in the country. All subjects were between 40 and 65 years old at the point of recruitment in 2008, and did not possess any pre-existing major health conditions, defined in this study to include cardiovascular disease, mental illness, diabetes, stroke, renal failure, hypertension and cancer. However, no detailed clinical assessments were performed to confirm the absence of above-stated diseases, and ascertainment depended on self-reports. The ethnic membership of each subject was assigned after verbal confirmation that all four grandparents belonged to the same ethnicity. All study subjects provided written informed consent for the participation, and all protocols associated with this study were approved by the National University of Singapore Institutional Review Board."

Major comment: Since the authors present their work as a "concept paper" it would be instructive to highlight why these selected "omics" platforms are providing a proper basis to compare ethnic subgroups.

- iii) The reviewer commented that it will be useful to describe why the selected omics platforms are providing a proper basis to compare between ethnic groups.

Our response:

We thank the reviewer for this comment to provide further basis to the selection of the omics platforms for the Integrative Omics Study. The selection of the omics technologies was made on the basis of the biological cascade from genetics to transcription to proteomics and miRNA, and then to expressed phenotypes such as lipidomics, metabolomics and microbiome, and observable traits and measurements such as the clinical parameters of HbA1c, triglycerides, cholesterol, etc. The Integrative Omics Study aims to study the value of physiological information at the different levels of DNA, mRNA and metabolites. The current manuscript has presented the preliminary data and findings from four omics platforms: genetics, transcriptomics, lipidomics and miRNA, and we are in the process of generating and applying for additional funding support to further generate baseline information for looking at the same samples at the protein level, metabolite level, gut and skin microbiomes, chip-based methylation epigenetics and to extend the transcriptomics to RNA-sequencing. Our investigations have already yielded important findings regarding the degree of population differentiation seen at each of the omics levels, as well as to identify at which level in the biological cascade did the biological measurements remained relatively conserved across the different ethnic groups. We agree with the reviewer that we should describe in the manuscript why the specific omics platforms were selected, and we have expanded the Introduction in the revised manuscript to include a brief explanation on the

choice of the omics technologies for the present setup. The specific paragraph in the Introduction reads as:

“In the present setup, measurements have been made in the iOmics to investigate the baseline genetics, transcription, lipid levels and microRNA expression or quantification. Each technology was selected for the purpose of evaluating the value of information in the biological cascade from DNA to RNA, and to biological units (cellular lipids) that are close surrogates to expressed phenotypes. The iOmics is expected to facilitate biomedical science experiments investigating the impact of an omic measurement on biological processes or outcomes, by interrogating the presence and extent of intra- and inter-omic correlation. In addition to the unprecedented collection of omic measurements made on the same individuals, the design and ethical set-up of the iOmics specifically offers the unique opportunity to recall participating subjects back for additional experiments according to the desired biological profiles. The data for the iOmics resource is publicly available at <http://phg.nus.edu.sg/#iomics>.”

Major comment: Large-scale RNA-sequencing to cover a higher fraction of all GENCODE genes as well as splice isoforms, and large-scale epigenetics measurements are notably missing and could potentially have identified more distinct regulatory differences between the ethnic groups

- iv) The reviewer noted that large-scale RNA-sequencing and large scale epigenetics measurements are missing and may have provided more information to identify regulatory differences between the ethnic groups.

Our response:

We are delighted that the reviewer has noted that the current study will benefit from including RNA-sequencing and the study of methylation differences between the ethnic groups. We agree entirely that the current set-up can benefit from further omic measurements, and we are pleased to report that we have secured funding and have commenced the experiments to measure methylation with the Illumina Infinium MethylationEPIC Kit for all the iOmics samples, as well as are in the process of starting the microbiome arm of the study to perform whole-genome sequencing of gut microbiota before and after commencing a dietary intervention for a subset of the iOmics subjects from all three ethnic groups. We are also in the process of applying for additional funding to expand transcriptomics measurements with RNAseq, and to include mass spectrometry metabolomics measurements from first-void urine from the iOmics subjects. The intention is for the iOmics Study to perform the deepest possible biological characterization and phenotyping, in order to generate a publicly available database across multiple omics technologies for subjects from three genetically distinct populations. This will facilitate the understanding of within-technology and between-technology co-variation. We have taken the decision to proceed with the present Concept manuscript as many of the current and future developments in the iOmics Study depend on the availability of funding, and we have received widespread interest in the use of our present datasets. We also believe there are already sufficient findings to warrant a first manuscript, and subsequent manuscripts can then report on the additional technologies or cross-technology correlations.

Minor comment: Methods: Where the blood samples for the lipidomics and transcriptomics taken after fasting and were they taken at the same time of the day for all individuals?

- v) The reviewer queried whether the blood samples for the lipidomics and transcriptomics assays were taken after fasting and were they taken at the same time of the day for all individuals.

Our response:

We clarify that the blood samples used for the lipidomics and transcriptomics measurements were taken post-fasting. We apologized that this was not specified clearly in the main text and we have revised the Materials and Methods section to explicitly state that “Blood sampling for the lipidomics and transcriptomics assays was performed after at least 12 hours of fasting”. While the recruitment of the iOmics subjects took place over a period of 6 months, the blood samples were always taken within the same period in the morning for all the participants, between 7am and 8am. This is to minimize any impact of unanticipated circadian patterns on underlying biological variations.

Minor comment: Discussion: miRNA is not an omics technology. Consider rewording the first sentence.

- vi) The reviewer suggested rephrasing the first sentence of the Discussion section as miRNA is not strictly an omics technology by itself.

Our response:

We thank the reviewer for this helpful suggestion to improve the clarity and accuracy of the descriptions. We have revised the manuscript as recommended by the reviewer, and to group the referencing of miRNA under transcriptomics.

Minor comment: Discussion: References missing in the discussion of the impact of this type of data infrastructure.

- vii) The reviewer commented that additional references can be included in the discussion on the impact and value of the different types of data infrastructure.

Our response:

We thank the reviewer for the recommendation, which we have taken on board and have included additional references that reflect the value of databases in providing scientific impact [REFs: 58-61], translational impact [REFs: 62-64] and implementation impact [REFs: 65]. In addition, we have also reorganized the first paragraph in the Introduction to provide specific illustrations of the different impact of baseline population databases for genetics, which reads:

“The subsequent development of national genome variation projects has thus produced numerous public databases that have been instrumental at enabling genetics as a forerunner in precision medicine⁹⁻¹². For instance, the predecessor of iOmics, the Singapore Genome Variation Project⁹ which only focused on making static genetic SNP and HLA measurements, indeed facilitated numerous investigations into the population genetics and genetics of common diseases in Asian communities, while at the same time allowed cost-effectiveness assessments and burden estimation of pharmacogenetic testing prior to initiate drug treatments^{13,14} which consequentially influenced policies on governmental subsidies for the costs of genetic tests¹⁵.”

Reviewer 3:

While the authors demonstrated the results for each measurement separately, they did not show the usefulness of integrating multi-omics datasets, mainly because of an insufficient sample size in the present study alone, as mentioned in the manuscript.

Even if the authors regard this as a concept paper, it is preferable (and more attractive) to the readers to show some examples for co-expression that exists between the different omics measurements and/or non-omic variables. For instance, they can pursue a mechanism underlying the observed transcriptomic difference at urotensin II between Indians and non-Indians, starting from the current set of 364 individuals to expanded, larger samples, as an example to utilize the omics towards integration

- i) The reviewer noted that while the manuscript has clearly described the results for each omics measurement separately, the present manuscript did not report on the usefulness for integrating across multiple omics. The reviewer suggested the possibility of providing some examples of co-expression that exists between the different omics measurements and/or non-omic variables.

Our response:

We thank the reviewer for this insightful comment, and for the positive endorsement of the value of the iOmics Study database. We are currently in the process of performing the cross-omic correlation analyses, specifically between genetics and gene expression, as well as genetics and lipidomics. For such analyses, there are substantial methodological developments necessary to mitigate the multiple-testing problem, as well as to reduce the impact of relatively small sample sizes (which again is also affected by the testing of multiple hypotheses).

For example, we have analysed the co-expression of genes in a network fashion and noticed how this affected the definition of cis-eQTLs and trans-eQTLs – where what appeared as a trans-eQTL is actually like due to a cis-effect to a gene within the same co-expression network. In this specific study, there is a need to carefully curate and define what a co-expression gene network is, as well as to perform an in-depth assessment of cis- and trans- effects. Another example of ongoing cross-omic developments that was made possible with the iOmic dataset is the apparent correlation of pathway-based genetic variation with lipidomics fluctuation in specific lipid classes. Here there is a need to properly describe how genetic pathways are built and to highlight how correlations between lipid species are utilized to reduce the number of hypotheses queried. In each of the two examples, the analyses and results rightfully warrant a separate report, which we will not be able to do justice if we attempt to include in the present Concept manuscript.

The iOmic Study has successfully brought together experts in genetics, transcriptomics, lipidomics and miRNA to jointly study a fixed group of volunteers from three ethnic populations in Singapore. As what was suggested by the reviewer, the resultant database has indeed sparked a number of unprecedented cross-omics investigations, especially in identifying those cross-omics correlations that are consistent and conserved across the different ethnic populations. We believe the sharing of these data in an open and timely fashion will facilitate the global community to extend their investigations using this data resource, rather than holding onto the data privately while we allow our internal collaborations to complete.

REVIEWERS' COMMENTS:

Reviewer #1 (Remarks to the Author):

The authors have thoughtfully addressed the critique and improved the manuscript readability. One small niggle, however, is that TG and DG, but also COH and CE are all neutral lipids. Hence in the context of Suppl Table 7 (and corresponding pieces in the main text) they are better categorized as glycerolipids, which should clearly distinguish them from COH and CE.

Otherwise the manuscript looks technically solid, very significant to the field and I am happy to endorse its publication

Reviewer #2 (Remarks to the Author):

Overall the authors' have provided in sufficient answers to all questions.

Regarding comment i/ The introductory paragraph has become a bit too broad and I suggest the authors remove or cut down on following part:

"The efficient design of GWAS for querying the entire genome benefitted from the arrival of the HapMap resource, which queried a large number of genetic markers in groups of individuals pseudo-randomly sampled from a few countries. The result of this endeavour was the production of a map outlining the correlation patterns in the human genome for identifying tagging single nucleotide polymorphisms (SNPs)^{4,5}. The HapMap resource also provided a public database on how prevalent specific alleles are in different ancestry groups in the world⁶. The ability to compare the extent of linkage disequilibrium (LD) differences between ancestry groups also meant index findings from GWAS can be evaluated for reproducibility across multiple population groups⁷⁻⁹."

Reviewer #3 (Remarks to the Author):

The authors' responses to the reviewer's comments seem to be appropriate. Among the points, which the authors indicated in their response letter, I would recommend the authors to describe in the Discussion section (presumably in the last paragraph) "the necessity of substantial methodological developments to mitigate the multiple-testing problem when people analyze the co-expression of genes using multi-omics datasets", in addition to the current cautionary notes about an insufficient sample size.

Reviewer 1:

The authors have thoughtfully addressed the critique and improved the manuscript readability. One small niggle, however, is that TG and DG, but also COH and CE are all neutral lipids. Hence in the context of Suppl Table 7 (and corresponding pieces in the main text) they are better categorized as glycerolipids,

which should clearly distinguish them from COH and CE. Otherwise the manuscript looks technically solid, very significant to the field and I am happy to endorse its publication

- i) The reviewer commented that the classification of DG and TG should be categorized as glycerolipids to better distinguish them from COH and CE.

Our response:

We thank the reviewer for the careful review of the manuscript and the recommendation that DG and TG should be categorized as glycerolipids. We have corrected our manuscript throughout to ensure that DG and TG are classified as glycerolipids. Specifically, Supplementary Table 4 (previously Supplementary Table 7) has been corrected, as well as in two corresponding statements in the main text which now read as:

“The set of 282 lipid species came from four lipid categories (glycerophospholipids, sphingolipids, sterol lipids, glycerolipids), of which there were 20 lipid classes (**Supplementary Table 4**).”

“Notably, lipid classes in the sterol (free cholesterol (COH), cholesteryl ester (CE) and glycerolipids (diglycerides (DG) and triglycerides (TG)) categories were highly correlated within and between classes, with lipids from DG and TG accounting for 22% (28/125), 16% (21/125) and 3% (2/65) of the observed lipid pairings.”

Reviewer 2:

Overall the authors' have provided in sufficient answers to all questions. Regarding comment i/ The introductory paragraph has become a bit too broad and I suggest the authors remove or cut down on following part:

“The efficient design of GWAS for querying the entire genome benefitted from the arrival of the HapMap resource, which queried a large number of genetic markers in groups of individuals pseudo-randomly sampled from a few countries. The result of this endeavour was the production of a map outlining the correlation patterns in the human genome for identifying tagging single nucleotide polymorphisms (SNPs)^{4,5}. The HapMap resource also provided a public database on how prevalent specific alleles are in different ancestry groups in the world⁶. The ability to compare the extent of linkage disequilibrium (LD) differences between ancestry groups also meant index findings from GWAS can be evaluated for reproducibility across multiple population groups⁷⁻⁹.”

- i) The reviewer recommended improving the brevity of the Introduction and recommended a section of the first paragraph in the Introduction to be removed or trimmed.

Our response:

We thank the reviewer for the recommendation to improve the brevity of the Introduction, which we have reviewed and made specific changes to the first paragraph without compromising on the key intent of discussing the HapMap resource, which now reads as:

“Knowledge of the genetic determinants of common human diseases has increased tremendously in the past decade, mostly from discoveries made by genome-wide association studies (GWAS)¹⁻³. The efficient design of GWAS for querying the entire genome benefitted from the arrival of the HapMap resource, which produced a genomic map that outlined the correlation patterns in the human genome for identifying tagging single nucleotide polymorphisms (SNPs)^{4,5}. The HapMap resource also provided a public database on how prevalent specific alleles are in different ancestry groups in the world⁶. The subsequent development of national genome variation projects has thus

produced numerous public databases that have been instrumental at enabling genetics as a forerunner in precision medicine⁹⁻¹². For instance, the predecessor of iOmics, the Singapore Genome Variation Project⁹ which only focused on making static genetic SNP and HLA measurements, indeed facilitated numerous investigations into the population genetics and genetics of common diseases in Asian communities, while at the same time allowed cost-effectiveness assessments and burden estimation of pharmacogenetic testing prior to initiate drug treatments^{13,14} which consequentially influenced policies on governmental subsidies for the costs of genetic tests¹⁵.”

Reviewer 3:

The authors’ responses to the reviewer’s comments seem to be appropriate. Among the points, which the authors indicated in their response letter, I would recommend the authors to describe in the Discussion section (presumably in the last paragraph) “the necessity of substantial methodological developments to mitigate the multiple-testing problem when people analyze the co-expression of genes using multi-omics datasets”, in addition to the current cautionary notes about an insufficient sample size.

- i) The reviewer recommended that the Discussion should highlight the need for substantial methodological developments to enable the analysis of multi-omics datasets.

Our response:

We thank the reviewer for this recommendation, which we strongly agree with. We have thus expanded the last paragraph of the Discussion to include a comment on the need for novel methodological work specifically designed to handle the multiple testing problem that accompanies any cross-omic analyses, particularly when genetics constitutes one of the cross-omic platforms. The last paragraph now reads:

“This paper has only scratched the surface on what the iOmics resource can deliver. Evidently there is considerable potential in the use of this dataset to investigate the degree of co-expression that exists between the different omics measurements. The relatively small sample size will undoubtedly hinder the discovery of networks with modest levels of co-expression. In addition, there is a dire need for novel methodologies to be designed with the specific intent of addressing the problem of multiple tests which invariably is present in such cross-omics analyses. But like the HapMap before this, identifying clear patterns of co-expression that are ubiquitously present across all three ethnic groups is a real possibility. The aspiration for the iOmics will be to integrate the present resource with longitudinal and prospective clinical records, where clinical decisions can be made not only to address clinical needs, but also with reference to the baseline omics, lifestyle and nutritional profiles.”